# Mixture Effects of Tryptophan Intestinal Microbial Metabolites on Aryl Hydrocarbon Receptor Activity

**DOI:** 10.3390/ijms231810825

**Published:** 2022-09-16

**Authors:** Aneta Vrzalová, Petra Pečinková, Peter Illés, Soňa Gurská, Petr Džubák, Martin Szotkowski, Marián Hajdúch, Sridhar Mani, Zdeněk Dvořák

**Affiliations:** 1Department of Cell Biology and Genetics, Faculty of Science, Palacký University, Šlechtitelů 27, 78371 Olomouc, Czech Republic; 2Institute of Molecular and Translational Medicine, Faculty of Medicine and Dentistry, Palacký University Olomouc, Hněvotínská 1333/5, 77900 Olomouc, Czech Republic; 3Department of Medicine, Molecular Pharmacology and Genetics, Albert Einstein College of Medicine, Bronx, NY 10461, USA

**Keywords:** aryl hydrocarbon receptor, tryptophan metabolites, indole derivatives, mimic mixtures, microbiome

## Abstract

Aryl hydrocarbon receptor (AHR) plays pivotal roles in intestinal physiology and pathophysiology. Intestinal AHR is activated by numerous dietary, endogenous, and microbial ligands. Whereas the effects of individual compounds on AHR are mostly known, the effects of real physiological mixtures occurring in the intestine have not been studied. Using reporter gene assays and RT-PCR, we evaluated the combinatorial effects (3520 combinations) of 11 microbial catabolites of tryptophan (MICTs) on AHR. We robustly (n = 30) determined the potencies and relative efficacies of single MICTs. Synergistic effects of MICT binary mixtures were observed between low- or medium-efficacy agonists, in particular for combinations of indole-3-propionate and indole-3-lactate. Combinations comprising highly efficacious agonists such as indole-3-pyruvate displayed rather antagonist effects, caused by saturation of the assay response. These synergistic effects were confirmed by RT-PCR as CYP1A1 mRNA expression. We also tested mimic multicomponent and binary mixtures of MICTs, prepared based on the metabolomic analyses of human feces and colonoscopy aspirates, respectively. In this case, AHR responsiveness did not correlate with type of diet or health status, and the indole concentrations in the mixtures were determinative of gross AHR activity. Future systematic research on the synergistic activation of AHR by microbial metabolites and other ligands is needed.

## 1. Introduction

The mutualistic microorganism community of the human gastrointestinal tract plays a crucial role in driving the development of the immune system and maintaining metabolic and tissue homeostasis. This is tremendously important to the well-being of the host since it represents a state without disruption, reactivity, or inflammation. All species interconnected in the gut generate a broad spectrum of metabolically active molecules from exogenous dietary substrates or endogenous host compounds to which the cells of the intestinal mucosa are exposed to and that have a significant impact on gut barrier function and immune responses [1,2]. Disruption of the host–microbiota interaction equilibrium has been implicated in multiple diseases, including inflammatory bowel disease (IBD), liver diseases, gastrointestinal cancers, metabolic diseases, and many others [3,4]. Therefore, the proper composition of these microbial metabolites is fundamentally crucial for human health. The identification of molecular targets for microbial metabolites helps to determine the mechanism of diseases and the development of potential therapeutics. A large number of microbial metabolites are derived from the gut microbiota-mediated metabolism of the essential amino acid *L*-tryptophan (*L*-Trp). Dietary tryptophan is metabolized by endogenous host cells (the kynurenine pathway and serotonin pathway) or intestinal microorganisms that convert unabsorbed *L*-Trp into indole (IND), tryptamine (TA), 3-methyl-indole (3MI, skatole), indole-3-aldehyde (IA), indole-3-acetate (IAA), indole-3-acetamide (IAD), indole-3-ethanol (IET), indole-3-pyruvate (IPY), indole-3-propionate (IPA), indole-3-acetaldehyde (IAL), indole-3-lactate (ILA), indole-3-acrylate (IAC), etc. [5,6]. Numerous microbial catabolites of tryptophan (MICTs) exhibit both agonist and antagonist activities at the aryl hydrocarbon receptor (AHR) [7,8,9,10], which is present in host intestinal epithelial cells and immune cells. During the past two decades, AHR has been characterized as one of the key sensors of metabolites produced by the intestinal microbiota, where it has been implicated in the regulation of the mucosal immune system and intestinal barrier function [6,11,12].

AHR is a cytosolic ligand-activated transcription factor that belongs to the basic helix-loop-helix (bHLH)/PerARNT-Sim (PAS) superfamily [13] and was originally identified as an environmental sensor of xenobiotic chemicals. Upon ligand binding, AHR translocates to the nucleus and heterodimerizes with the AHR nuclear translocator (ARNT/HIF1β) and promotes the transcription of multiple responsive genes harboring AHR-responsive DNA elements (DRE—dioxin response element or XRE—xenobiotic response element) in their regulatory regions, e.g., *CYP1A1*, *CYP1A2*, and *CYP1B1* [14,15,16] or *AHRR* [17]. Beyond xenobiotic metabolism, it has been recognized that AHR regulates diverse genes such as *p27^KIP1^, p21^CIP1^, c-jun, junD, IL-6, IL-22, Bax, PAI-1,* and many others [18], suggesting AHR has broad implications in many physiological and pathophysiological processes [19,20,21,22,23]. Therefore, AHR is an emerging molecular target for the development of new therapeutics. The promiscuous ligand-binding domain (LBD) of AHR binds a number of xenobiotics (drugs, environmental pollutants, dietary compounds) and multiple physiologic ligands that are produced by the host organs and commensal flora. In comparison with bulky polycyclic planar molecules, which cause persistent activation of AHR, endogenous and microbial AHR ligands are significantly smaller in size, and they often have moderate to low affinity for AHR because they interact with different residues of the AHR LBD than high-affinity ligands such as dioxin [24,25]. Such lower endogenous levels of activation of AHR have been shown to be beneficial in the maintenance of immune health and intestinal homeostasis [9].

In our recent study, we demonstrated that a series of MICTs are ligands and agonists of AHR [10]. Whereas IND, IAD, and IPY are high-efficacy AHR agonists, IET, IAC, 3MI, and TA have medium efficacy, and IPA, IAA, ILA, and IA have no or low efficacy. It is worth noting that results from the reporter hepatic cell line were obtained after 24 h of incubation; therefore, the outcomes may be influenced by the active hepatic metabolism of tested compounds. Indeed, while we observed EC_50_ of about 1500 µM for indole, a study from Perdew’s lab reported EC_50_ of around 3 µM after 4 h of incubation in human hepatic HepG2 (40/6) reporter cells with indole [8]. In addition, we reported differential AHR relative efficacies of series of methylindoles and methoxyindoles when applied in short versus long time periods [26]. Hence, the first objective of the current study was to characterize in detail the AHR agonist effects of MICT series (11 compounds) using reporter gene assay and a 4-h incubation time. This approach allowed an expansion of existing data to obtain a solid background to interpret data from short- and long-term incubations.

Human intestinal microbiota produces a wide array of AHR-active microbial metabolites, with the most promiscuous being MICTs and short-chain fatty acids (acetate, propionate, butyrate). In addition, a large number of dietary (e.g., indole-3-carbinol condensation products) and endogenous (e.g., kynurenic acid) AHR ligands are present in the intestines. Therefore, intestinal AHR is continuously exposed to a multicomponent mixture of AHR ligands. In this context, knowledge of the effects of single individual compounds on AHR cannot be trivially translated to predict the effects of mixture. Studies on the mixture effects of MICTs are limited to those reporting the enhancement of AHR responsiveness to tryptamine and indole by short-chain fatty acids [27]. Several studies reported that butyrate augments ligand-inducible expression of AHR target genes via inhibition of histone deacetylase catalytic activity. Beyond this fact, we have recently elucidated in detail the mechanism of the synergistic effects of AHR-agonist between butyrate and microbial metabolite FICZ (6-Formylindolo [3,2-*b*] carbazole). We showed that butyrate does not bind to AHR and does not enhance FICZ-mediated nuclear translocation of AHR and formation of the AHR-ARNT heterodimer. On the other hand, butyrate increased the enrichment of *CYP1A1* promoter with FICZ-activated AHR, and increased the FICZ-induced expression of *AHR* target genes [28]. Hence, the second objective of this study was to describe the mixture effects of MICTs at AHR and to elucidate the underlying principles defining the difference between the isolated and combined effects of MICTs.

In the present work, we studied the mixture effects of MICTs at AHR using two approaches: (i) Examination of binary mixtures of MICTs using a high-throughput platform and a stably transfected reporter gene cell line. We identified MICT combinations with synergistic effects at AHR, which were further confirmed at the level of *CYP1A1* mRNA expression; and (ii) testing of mimic mixtures of MICTs, which were prepared based on the metabolomic data from human feces and colonoscopy aspirates. We concluded that the overall effects of mixtures at AHR correlated with the indole levels in these mixtures.

## 2. Results

### 2.1. Cell Viability Assay and Nuclear Receptor Counter-Screen of MICTs

Prior to the gene reporter assay, the effects of MICTs on the viability of the LS174T and AZ-AHR cell lines were evaluated using the MTT, neutral red uptake, and LDH leakage assays (Figure 1A). All compounds were tested at concentrations of up to 200 µM, except for IAC (the maximum concentration was 100 µM due to solubility limitations) and IND (1 mM given its known high intestinal concentrations) [29,30]. The majority of the entire series of MICTs did not significantly alter cell survival, causing less than a 20% drop in cell viability (Figure 1A). Indole was toxic at very high concentrations (2.5 and 5 mM).

We performed a counter-screen for the potential ability of MICTs to activate a panel of nuclear receptors, including vitamin D receptor (VDR), glucocorticoid receptor (GR), peroxisome proliferator-activated receptor gamma (PPARγ), and androgen receptor (AR). Stably transfected reporter cell lines were incubated for 24 h with MICTs, vehicle DMSO (0.1% *v*/*v*), and model agonists of each nuclear receptor VD3 (75 nM; fold induction 65×), DEX (100 nM; fold induction 22×), 15d-PGDJ (40 µM; fold induction 3×), or DHT (100 nM; fold induction 20×). The tested MICTs did not activate any of the examined receptors (Figure 1B), implying that these receptors are not off-targets for MICTs. On the other hand, the activation of pregnane X receptor (PXR) by several MICTs was recently reported [31].

### 2.2. High-Throughput Screening (HTS) of AHR Receptor Activity by Individual MICTs

We evaluated the agonistic effects of individual MICTs on AHR in a stably transfected reporter human hepatoma cell line AZ-AHR after 4 h of treatment. The relative efficacy and half-maximal effective concentration (potency; EC_50_) of each compound were determined (Figure 2). Each MICT activated AHR in a dose-dependent manner (Figure 2, Panel B) and the relative efficacies and potencies of individual MICTs differed substantially (Figure 2, Panel A). The highest potency was observed for IAD, IAC, 3MI, and TA with EC_50_ of 4.5, 8.4, 8.4, and 13.8 µM, respectively. The most efficacious agonists of AHR were IAD, IPY, IAC, and IND, displaying relative efficacies of 84%, 77%, 68%, and 66%, respectively, as compared to 5 nM TCDD (luciferase induction ≈200-fold). 3MI, IAA, IET, ILA, and TA displayed medium-efficacy agonism (≈50%). Agonists with the lowest efficacy were IA and IPA (≈25%).

### 2.3. High-Throughput Screening (HTS) of AHR Receptor Activity by Binary Mixtures of MICTs

The combination effects of binary mixtures of MICTs at AHR were evaluated using the gene reporter assay in a high-throughput set-up. AZ-AHR cells were incubated for 4 h with vehicle, TCDD, and binary mixtures of MICTs. Within the tested series of MICTs (11 compounds), each MICT was combined with another one, thereby giving 55 individual pairs of compounds. For each pair of MICTs, both components were applied in 8 different concentrations, thereby yielding 64 individual combinations. Overall, 3520 individual binary MICT mixtures were examined. The nature of the interactions (synergistic, additive, or antagonistic) is represented as a logarithm of combination index (CI) and is color coded in the heat-map shown in Figure 3 (green indicates high synergy). Synergistic effects located in a range of log 0.3 to 0.7 were observed mainly between low- or medium-efficacy agonists, including: 3MI + ILA, IA + ILA, ILA + TA, ILA + IPA, IA + IPA, IET + IPA, IPA + IPY, IPA + TA, IPY + IA, and IPY + TA. Binary mixtures containing high-efficacy agonist (IAD, IPY, IAC, IND) displayed a rather antagonistic effect (Figure 3).

### 2.4. Binary Mixtures of ILA and IPA Synergistically Induce CYP1A1

Based on the results of HTS in the reporter gene assay, we selected candidate MICT binary mixtures with the highest synergistic effects at AHR and tested whether these synergistic effects also occurred at the level of endogenously expressed CYP1A1, a prototypical AHR-target gene. For this purpose, AZ-AHR cells were incubated for 4 h with vehicle, TCDD, and candidate binary mixtures. All tested compounds and mixtures induced *CYP1A1* mRNA (Figure 4), and the relative induction correlated well with the results obtained by the reporter gene assay (Figure 2 and Figure 3). This synergism was classified into two types: (i) Type I, where the efficacy of the binary mixture is higher than the effect of each compound alone (Figure 4A). This type of synergism was observed for combinations: IPA 66.7 µM + ILA 66.7 µM, IPA 66.7 µM + TA 7.4 µM, IPA 200 µM + TA 7.4 µM, and ILA 66.7 µM + TA 7.4 µM; and (ii) Type II, where the efficacy of the binary mixture is higher than the sum of the efficacies of individual compounds alone (Figure 4B). This synergism occurred in the following mixtures: IPA 66.7 µM + IA 7.4 µM, IPA 66.7 µM + IET 0.82 µM, IPA 66.7 µM + IPY 0.091 µM, and IPY 0.091 µM + IA 7.4 µM.

### 2.5. Effect of Multicomponent Mimic Fecal Mixtures of MICTs on the AHR Activity

In a recent study [30], the levels of AHR-active MICTs were quantified in human fecal samples. A study was performed in subjects on a defined diet (stool samples from a controlled clinical nutrition diet study [32]) and subjects with unrestricted (ad libitum) dietary consumption. We prepared multicomponent mimic mixtures of MICTs, containing IND, TA, IA, IAA IAC, 3MI, and serotonin (SER), using the concentrations of individual MICTs (mean value) reported in the mentioned study [30]. We examined the activation of AHR by both mimic fecal mixtures (defined and unrestricted diet) and by their individual components using the stably transfected reporter intestinal cell line LS174T-AHR-luc (Figure 5). The relative efficacy attained by IND was approximately 123% (IND; 199 µM) and 130% (IND; 486 µM), which indicates saturation of AHR activity occurred. The activation of AHR by both mimic mixtures was approximately 150% of the TCDD-induced activity and did not differ significantly from the AHR effects of IND itself. These data imply there is no synergism between individual MICTs contained in mimic fecal mixtures, and that indole accounts for the majority of the AHR agonist effects of these mixtures.

### 2.6. Effect of IND and IPA Binary Mimic Mixtures on AHR Activity

In our recent study [33], we quantified the levels of IND and IPA in serial colonoscopy aspirates from healthy volunteers and patients suffering from intestinal inflammatory diseases. Based on these data, we prepared mimic binary mixtures of IND and IPA, and tested them for AHR-agonist activity in the LS174T-AHR-luc reporter cell line. The levels of IND and IPA were highly variable and did not differ significantly between healthy subjects and IBD patients (Figure 6 middle and bottom). The highest applied concentration of IND (274 µM) activated AHR stronger than TCDD. Moreover, the presence of indole at this concentration provided maximal AHR activation regardless of the presence of IPA (IPA_max_ 19.5 µM), as seen from samples P25 and P14 (Figure 6 top). AHR activation by mimic binary mixtures remained on a similar level with decreasing concentrations of indole (Figure 6 top), reflecting that AHR was saturated under higher indole concentrations (>20 µM), as shown in the dose–response experiments (Figure 2). Consistently, AHR activation by mimic binary mixtures containing low levels of indole was proportional to the concentration of indole in these mixtures. Altogether, we did not observe synergism between IND and IPA in mimic binary mixtures under these particular experimental conditions.

## 3. Discussion

In recent years, the effects of microbial catabolites of tryptophan (MICTs) on AHR have been extensively studied in the context of gastrointestinal health and disease [8,10,26,34,35]. The majority of studies investigated single-molecule effects at AHR, yielding valuable and convincing data on the roles of MICT-activated AHR in human intestinal physiology and pathology. However, the human intestine is continuously exposed to a complex mixture of AHR-active compounds, including microbial metabolites, drugs, environmental pollutants, and dietary compounds. Whereas the effects of single molecules at AHR may be interpreted simply and straightforwardly, the mixture effects involve synergistic, additive, and antagonist interactions between the individual constituents.

Here, we studied the mixture effects of MICTs at AHR using two strategies: (i) We examined binary mixtures of MICTs (11 compounds) using a high-throughput platform (3520 combinations) and a stably transfected reporter gene cell line. We robustly (n = 30) determined the potencies (EC_50_ values) and relative efficacies of single MICTs. Based on the magnitude of luciferase induction, the tested MICTs were sorted out into high (IAD, IPY, IAC, IND), medium (3MI, IAA, IET, ILA, TA), and low (IA, IPA) efficacy AHR agonists. The most potent MICTs were IAD, IAC, 3MI, and TA. Synergistic effects of MICT binary mixtures were observed between low- and medium-efficacy agonists. The strongest synergistic effects were shown for combinations of IPA or ILA. On the other hand, binary MICT combinations comprising highly efficacious agonists such as IND or IPY displayed rather antagonist effects. This behavior might be explained by saturation of the assay response when combining two highly efficacious agonists that can reach a maximal 100% efficacy, which is an arbitrary ceiling. (ii) We tested mimic multicomponent mixtures of MICTs in physiological concentrations, which were prepared based on metabolomic analyses of human feces from individuals on a defined and unrestricted diet. The most abundant MICT in human stool was indole, which activated AHR to a similar extent to TCDD. The activity of mimic multicomponent mixtures at AHR, regardless of the type of diet, correlated with the effects of indole. This implies an apparent antagonist effect of the mixture due to the saturation of the biological response (vide supra). We also tested binary mimic mixtures of IND + IPA based on the concentration of these MICTs in colonoscopy aspirates from healthy individuals and IBD patients. We concluded that AHR responsiveness did not correlate with health status and that the indole concentrations in the mixtures are determinative of the gross AHR activity of the mixtures.

Synergistic effects of two or more compounds at AHR may occur through four mechanisms: (i) *positive allosteric modulation*: This is typical for many receptors, where one ligand binds a conventional ligand-binding pocket (orthosteric), and the second ligand binds allosterically, thereby positively modulating (enhancing) the receptor’s response. So far, allosteric ligands for AHR have not been described. (ii) *Supramolecular ligands*: This phenomenon was demonstrated for pregnane X receptor (PXR), which has a spacious ligand-binding pocket, allowing for distinct binding of diverse ligands. Bourguet´s group described synergistic activation of PXR by a binary mixture of polychlorinated biphenyl and estrogen hormone. Whereas individual compounds were PXR-inactive, they bound in the PXR ligand-binding pocket simultaneously in such a way that they enhanced their affinity for each other [36]. In the case of MICTs, it was shown that co-occupancy of indole and IPA at the human PXR ligand-binding pocket is feasible [37]. The ligand-binding pocket of AHR is also large [38] and predisposed to potentially accommodate several molecules simultaneously. In silico modeling and docking provided evidence of possible adaptation of the human AHR to bind two molecules of indole through a unique bimolecular mechanism [8]. Hence, a mechanistic explanation of MICTs’ synergistic effects at AHR through formation of supramolecular ligand is plausible. (iii) *Off-target*: It is common for chemicals to have more than one cellular target, and that conventional orthosteric binding and activation of the receptor by one compound may be modulated by interaction of the second compound with another target. Sensu stricto, two MICTs may bind both target (AHR) and off-target. This example can result in dual activation of AHR and PXR by the common ligand when mutual transcriptional and functional cross-talk between AHR and PXR is well-established. Indeed, we recently described activation of PXR by several MICTs [31]. Another example is the inhibition of histone deacetylase by butyrate, which enhanced AHR responsiveness to multiple ligands, including microbial ones [27,28]. Interaction of MICTs with other putative off-targets as a basis for MICTs’ synergistic effects may be the case. (iv) *Pharmacokinetic interaction*: The interaction between a ligand and a receptor within the cell is fundamentally influenced by a ligand pharmacokinetic, involving ligand metabolism, cellular uptake, and efflux. Pharmacokinetic interaction may be considered as a special case of off-target. For instance, the highly potent AHR ligand FICZ is rapidly metabolized and inactivated by CYP1A1 enzyme. Co-incubation of cells with FICZ and CYP1A1 inhibitor (e.g., ketoconazole) resulted in drastically enhanced AHR activation by FICZ [39]. This scenario could apply for MICTs, because we recently described that selected monomethylated indoles are extensively metabolized in cell cultures and that they are inhibitors of CYP1A1 catalytic activity [40].

In the current work, we described for the first time the mixture effects of microbial intestinal catabolites of tryptophan on AHR transcriptional activity. In future work, systematic research on the synergistic activation of AHR and other sensors for microbial ligands, such as PXR, by microbial metabolites and other ligands needs to be performed. Both the mechanistic aspects and biological relevance of mixture effects deserve future attention.

## 4. Materials and Methods

### 4.1. Chemicals and Reagents

Indole (IND), 3-methylindole (3MI), indole-3-acetate (IAA), indole-3-acrylate (IAC), indole-3-acetamide (IAD), indole-3-lactate (ILA), indole-3-propionate (IPA), indole-3-pyruvate (IPY), tryptamine (TA), and cholecalciferol (VD3) were obtained from Santa Cruz Biotechnology (Santa Cruz, CA, USA). Indole-3-aldehyde (IA), indole-3-ethanol (IET), indole-3-carboxylate (ICA), serotonin (SER), dimethylsulfoxide (DMSO), Triton X-100, dexamethasone (DEX), 15-Deoxy-δ12,14-Prostaglandin J2 (15d-PGDJ), 5α-dihydrotestosterone (DHT), thiazolyl blue tetrazolium bromide (MTT), McCoy’s 5A medium, Dulbecco’s modified Eagle’s medium (DMEM), RPMI-1640 medium, fetal bovine serum, charcoal-stripped fetal bovine serum, and hygromycin B were obtained from Sigma-Aldrich (St. Louis, MO, USA). 2,3,7,8-tetrachlorodibenzo-p-dioxin (TCDD) was acquired from UltraScientific (North Kingstown, RI, USA). Luciferase Assay System, Nano-Glo Luciferase AssaySystem, and Reporter Lysis 5 × Buffer were purchased from Promega (Madison, WI, USA). All compounds used for treatments were of ≥98% purity and other chemicals were of the highest commercially available quality.

### 4.2. Cell Cultures

The human Caucasian colon adenocarcinoma cell line LS174T was purchased from the American Type Culture Collection (ATCC). The following stably transfected reporter gene cell lines were used to estimate specific receptor activation: AZ-AHR (derived from the human hepatocellular carcinoma cell line HepG2) [41], LS174T-AHR-luc (harboring the reporter plasmid pGL-4.27-DRE and prepared by the same protocol as AZ-AHR, AZ-GR (derived from the human cervix carcinoma cell line HeLa)) [42], and IZ-VDRE (derived from the human colon adenocarcinoma cell line LS180) [43]. Cells were cultured in Dulbecco’s modified Eagle’s medium supplemented with 10% fetal bovine serum, 4 mM L-glutamine, 1% nonessential amino acids and antibiotics, and 100 U/mL Penicillin-Streptomycin. AIZ-AR (derived from the human prostate carcinoma cell line 22RV1) [44] was cultivated in RPMI-1640 medium supplemented with 10% charcoal-stripped fetal bovine serum, 4 mM L-glutamine, and antibiotics 100 U/mL Penicillin-Streptomycin. PAZ-PPARg (derived from the human bladder carcinoma cell line T24/83) [45] was cultivated in Dulbecco’s modified Eagle’s medium supplemented with 10% charcoal-stripped fetal bovine serum, 4 mM L-glutamine and antibiotics, and 100 U/mL Penicillin-Streptomycin. Cells were maintained in 75 cm^2^ plastic tissue culture flasks and subcultured twice a week. After every 3rd passage, 0.2 mg/mL of hygromycin B (10687010, Invitrogen) was added to the growth medium. Cells were incubated at 37 °C in the 5% CO_2_ atmosphere of a humidified incubator.

### 4.3. Human Studies

The colonoscopy aspirates were collected under an approved study (#2015-4465; audited by the IRB on 24 April 2019; NCT0408950). Specimens were obtained from unselected patients with inflammatory bowel disease (acute and chronic, Crohn’s (n = 9) and ulcerative colitis; (n = 6)) (mean age 52 years, 8 males: 7 females) and non-inflammatory bowel disease patients seen in the IBD clinic (n = 21; mean age 57 years, 10 males; 11 female). The concentrations of indole and IPA were determined by gas chromatography and mass spectrometric (GC-MS) analysis. Details of the clinical study (NCT04089501) and sample analysis were published elsewhere [33]. The levels of IND and IPA were normalized to the total aspirate protein concentrations.

### 4.4. Cell Viability Assays—MTT, Neutral Red, and LDH

LS174T and AZ-AHR cells were seeded into 96-well plates at a density of 4 × 10^4^ in 0.2 mL of the culture medium. After 24 h, the cells were treated for 24 h with the studied compounds in concentrations ranging from 1 nM to 200 µM (indole from 10 µM to 10 mM). The highest concentrations of the tested compounds were adjusted based on their solubility. The vehicle DMSO (0.1% *v*/*v*) was used as a negative control and Triton X-100 as a positive control. The MTT (3-(4,5,-dimethylthiazol-2-yl)-2,5-diphenyltetrazolium bromide) colorimetric assay was performed as previously described [41]. The neutral red assay and LDH (Abcam, Cambridge, UK) assays were performed according to the manufacturer’s protocol. Assays were measured spectrophotometrically using a Tecan Infinite M200 plate reader (Tecan, Männedorf, Switzerland).

### 4.5. Luciferase Reporter Assay

The reporter cells were seeded into 96-well plates at a density of 2.5 × 10^4^ (AZ-AHR, LS174T-AHR-luc, IZ-VDRE), 5 × 10^4^ (AIZ-AR), 2.2 × 10^4^ (AZ-GR), or 4 × 10^4^ (PAZ-PPARg) cells in 0.2 mL of culture medium. Following 16 h of stabilization, cells were treated with the tested compounds as described in the figure captions for the indicated period of time. The vehicle DMSO (0.1% *v*/*v*) was used as a negative control. After treatment, cells were lysed in a lysis buffer and the luciferase activity was measured in a 96-well plate format using the Nano-Glo Luciferase AssaySystem for the PAZ-PPARg and IZ-VDRE cell lines and the Luciferase AssaySystem for the AZ-AHR, LS174T-AHR-luc, AIZ-AR, and AZ-GR cell lines on a Tecan Infinite M200 Pro Plate Reader (Tecan, Männedorf, Switzerland).

### 4.6. High-Throughput AhR Reporter Gene Screening Assay

To assess the enhanced effect of combined treatments in the synergy experiments, the AZ-AHR reporter cell line was seeded into 384-well white CulturePlates (6007689, PerkinElmer, Waltham, MA, USA) at a density of 4 × 10^3^ cells per well by the dispenser Multidrop Combi (ThermoFisher Scientific, Waltham, MA, USA). After 24 h of incubation, the cells were treated with the reference (5 nM TCDD) and tested compounds using a contact-free acoustic liquid handler ECHO 550 (Labcyte, San Jose, CA, USA). All tested compounds alone or in binary combinations were analyzed at a concentration range of 0.091–200 µM except IAC (0.046–100 µM) and indole (0.46–1000 µM). The cells were incubated with the tested compounds for 4 h. Thereafter, the cells were washed with PBS by Washer EL406 (BioTek, Winooski, VT, USA) and lysed by 1x gene reporter lysis buffer (E3971, Promega). After adding the substrate (D-Luciferin, L9504, Sigma) with a dispenser Multidrop Combi (ThermoFisher Scientific), the luciferase activity was measured on a multimode plate reader EnVision (PerkinElmer). The experiments were performed in technical duplicates and three biological replicates at least.

Using the fold induction values, the half-maximal effective concentration values (EC_50_) and relative efficacy for each compound were calculated. The effects of binary mixtures were assessed using the CalcuSyn software (version 2.0; Biosoft, Novosibirsk, Russia) as a combination index (CI) and the Chou-Talalay method [46,47]. To assess the enhanced effect of the combined treatments in the synergy experiments, the effects were expressed as the fraction affected (FA) normalized to cells treated with TCDD and was calculated using the following equation: FA = sample/PC control, where the sample is the average RLU values from the wells treated with the tested compounds, and the PC control is the average RLU of the cells treated with TCDD.

### 4.7. Isolation of RNA and Quantitative Reverse Transcription Polymerase Chain Reaction (qRT-PCR)

AZ-AHR cells were seeded into 6-well plates at a density of 10^6^ in 2 mL of culture medium. Following 16 h of stabilization, the cells were treated with the tested compounds as described in the figure captions, vehicle (DMSO, 0.1% *v*/*v*), or TCDD (20 nM) for 4 h. The total RNA was isolated by TRI Reagent^®^, followed by reverse transcription using M-MuLV Reverse Transcriptase. The KiCqStart Probe Assay (Sigma Aldrich, Prague, Czech Republic) was used to determine the levels of CYP1A1 and GAPDH mRNA, and analysis was conducted using the Light Cycler 480 II apparatus (Roche Diagnostic Corporation, Prague, Czech Republic). Measurements were carried out in triplicate. Gene expression was normalized to GAPDH as a housekeeping gene. The data were processed by the delta-delta method.

### 4.8. Statistical Analysis

A student t-test, one-way analysis of variance (ANOVA), and calculations of EC_50_ values were performed using GraphPad Prism version 9.0 for Windows (GraphPad Software, La Jolla, CA, USA).

## Figures and Tables

**Figure 1 ijms-23-10825-f001:**
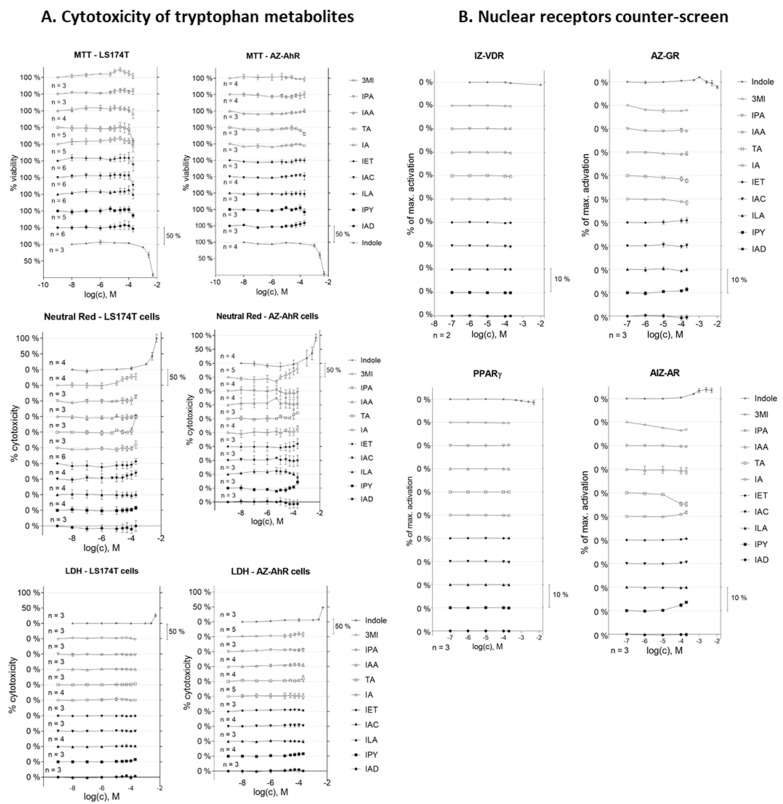
The effect of tryptophan metabolites on the cell viability and the activity of nuclear receptors. Cells were treated for 24 h with the tested compounds at concentrations ranging from 1 nM to 200 µM (indole from 10 µM to 10 mM). (**A**) Cell viability in LS174T and AZ-AHR cells. The MTT, NR, and LDH assays were performed as described in the experimental section. Data are the means from at least three consecutive cell passages and are expressed as a percentage of the viability (MTT) or cytotoxicity (NR, LDH) of the control cells. (**B**) The activity of VDR, GR, PPARγ, and AR receptors in stably transfected reporter cell lines. A luciferase assay was performed as described in the experimental section. Vehicle DMSO (0.1% *v*/*v*) was used as a negative control, and VD3 (75 nM), DEX (100 nM), 15d-PGDJ (40 µM), or DHT (100 nM) as positive controls, respectively. Data are the means from at least three consecutive cell passages and the effects of tested compounds are expressed as a % of induction by the positive control.

**Figure 2 ijms-23-10825-f002:**
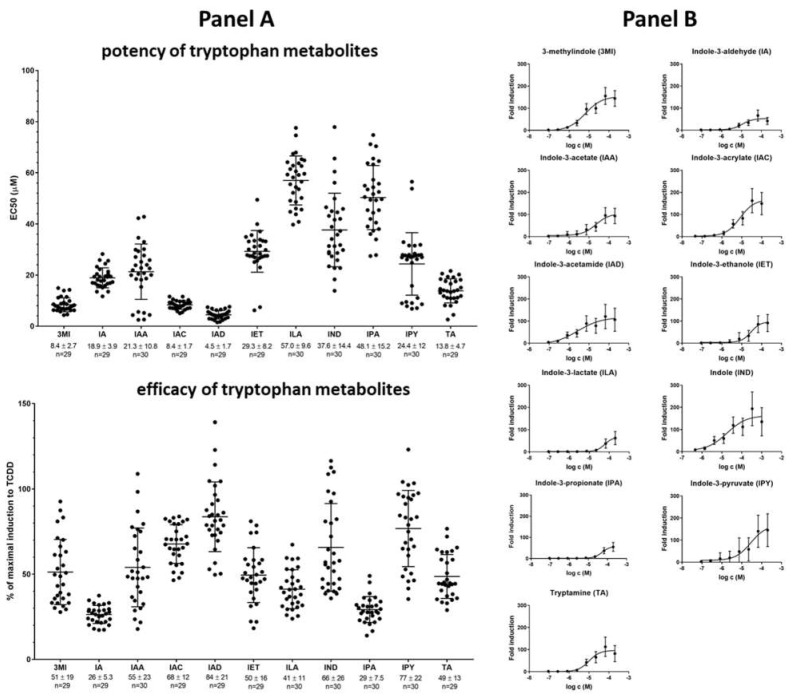
High-throughput screening of AHR transcriptional activity by tryptophan microbial catabolites (MICT). AZ-AHR cells were incubated for 4 h with vehicle (DMSO; 0.1% *v*/*v*), TCDD (5 nM), or increasing concentrations of the tested compounds. Following the treatments, cells were lysed, and luciferase activity was measured. Panel (**A**): The column scatter plot represents the values of the half-maximal effective concentrations (potency; EC_50_) and relative efficacies (a ratio of luciferase activity by MICT in highest concentration/luciferase activity by TCDD). Panel (**B**)**:** Dose–response assessment of AHR-dependent activity. Data are expressed as the fold induction of luciferase activity over control cells and are the mean ± SD from approximately 30 consecutive cell passages. The fold induction of TCDD was 200 ± 105 (n = 165). All values are statistically significant (*p* < 0.01) in comparison to vehicle-treated cells.

**Figure 3 ijms-23-10825-f003:**
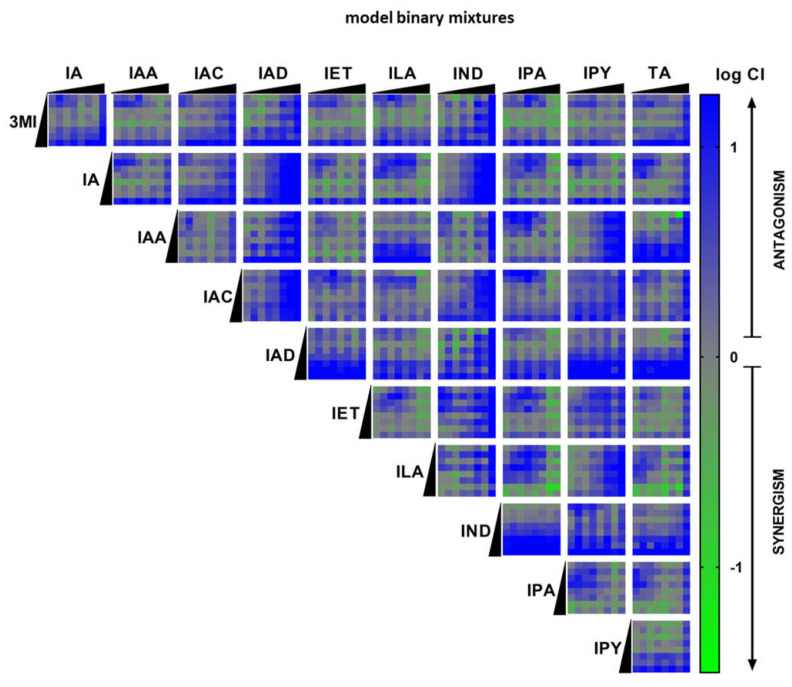
Heat-map of combination index (CI) values from high-throughput screening of the MICT binary mixtures at the AHR receptor. AZ-AHR gene reporter cells were incubated for 4 h with vehicle (DMSO; 0.1% *v*/*v*), TCDD as a positive control, and increasing concentrations of MICTs alone or as binary mixtures, one with each other in the entire concentration range. Following the treatments, cells were lysed, and luciferase activity was measured as described in the experimental section. The experiments were performed in technical duplicates and three biological replicates at least. The effect of MICTs in binary mixtures on the AHR transcriptional activity was analyzed by calculations based on the Chou-Talalay method and using the CalcuSyn software. The effects of the combined treatments are represented as a combination index (CI) and are color coded (green indicates high synergy and blue indicates no synergy).

**Figure 4 ijms-23-10825-f004:**
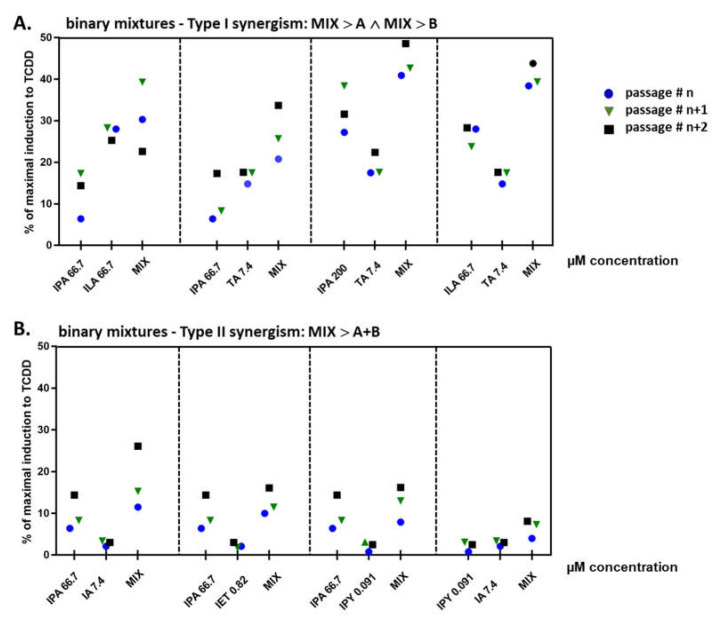
The induction of *CYP1A1* by selected MICTs and their binary combinations. (**A**) Type I synergism, where the efficacy of the binary mixture is higher than the effect of each compound alone; (**B**) Type II synergism, where the efficacy of the binary mixture is higher than the sum of the efficacies of two single compounds. AZ-AHR cells were incubated with vehicle (DMSO; 0.2% *v*/*v*), TCDD (20 nM), or MICTs alone or in binary combinations for 4 h. The levels of *CYP1A1* mRNA were determined by qRT-PCR and are expressed as a percent of induction by TCDD. Each dot represents one independent experiment. The experiments were performed in technical triplicates and three biological replicates at least.

**Figure 5 ijms-23-10825-f005:**
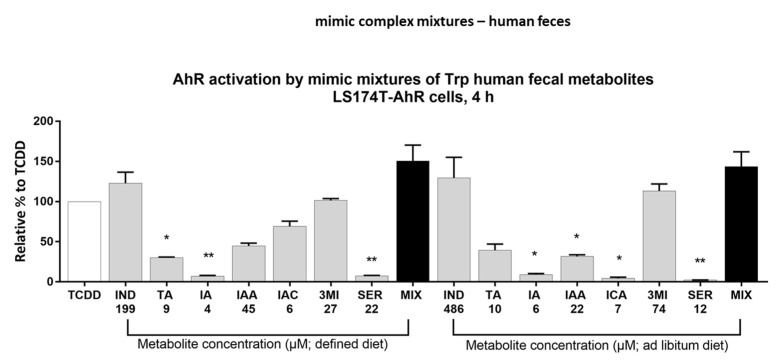
Effect of mimic mixtures of MICTs on AHR activity. LS174T-AHR-luc cells were incubated for 4 h with vehicle (DMSO; 0.2% *v*/*v*), TCDD (20 nM), and MICTs alone or in the mimic mixture. The concentration of each compound equals the observed mean quantified in human fecal samples [30]. The effects of tested compounds are expressed as a percent of induction of luciferase activity by TCDD over control cells. The experiments were performed in technical quadruplicates and three biological replicates. The error bars represent the mean ± SD, * *p* < 0.1, ** *p* < 0.01.

**Figure 6 ijms-23-10825-f006:**
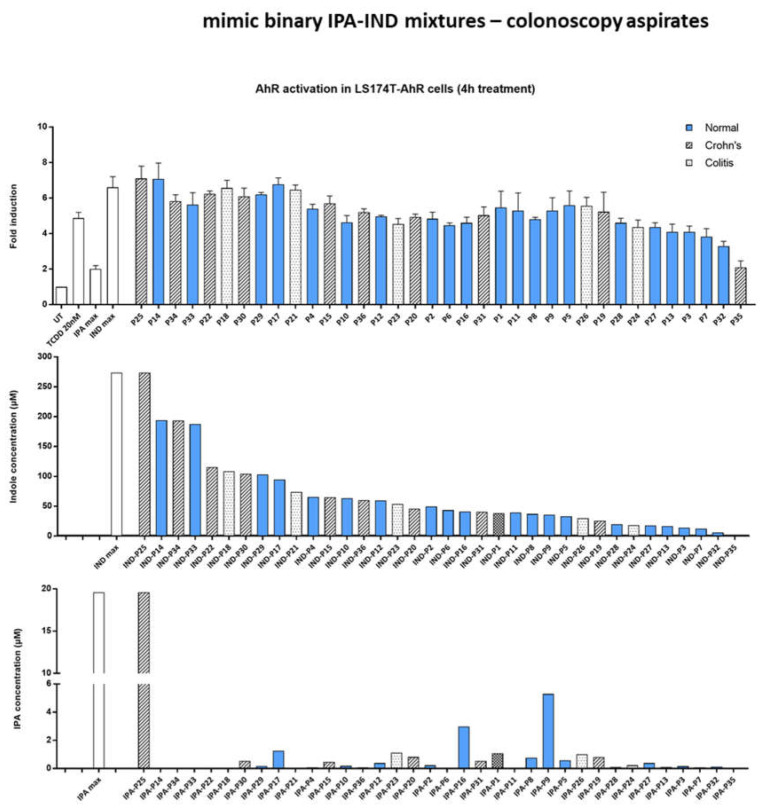
AHR activation by mimic binary mixtures of IND and IPA. LS174T-AHR-luc cells were incubated for 4 h with vehicle (DMSO; 0.2% *v*/*v*), TCDD (20 nM), and the tested compounds alone or in the mixture. Mimic binary mixtures were prepared according to the ratio of IND and IPA that equals the concentration detected in individual aspirates of colonoscopy patients. The experiments were performed in technical quadruplicates and three biological replicates.

## Data Availability

The data presented in this study are available online in this article.

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
