# Peer review of "Mixture Effects of Tryptophan Intestinal Microbial Metabolites on Aryl Hydrocarbon Receptor Activity"

_ijms, 2022, doi:10.3390/ijms231810825_

Round 1
Reviewer 1 Report
Aneta Vrzalova and coauthors investigated the effects of 11 tryptophan intestinal microbial metabolites (MICTs) on the aryl hydrocarbon receptor (AHR) activity. They analyzed the effect of individual, binary and mimic multicomponent mixtures of MICTs in order to get closer to the physiological human state.
The authors identified synergistic as well as antagonistic effects of binary MICTs combinations on AHR activity which can be explained by a saturation of assay response. They also highlighted a determinative role of indole concentrations for gross AHR activity.
Overall, the manuscript is clear, relevant for the field and presented in a well-structured manner.
However, a series of issues need to be addressed:
Major issues:
- It is not completely clear why the authors decided to analyze the effect of these 11 specific MICTs. I supposed that it is based on their recent study (Vyhlidalova et al., Trends Pharmacol Sci 2020) as outlined in the introduction section (line 79-89). The authors should clearly reformulate this part.
- Which cell line was used in Perdew’s lab study (line 85)? Is it a hepatic cell line as well? The authors should add this information as they set the incubation time regarding of this study.
- The figure with the negative and positive controls for the luciferase assay should be added as a supplemental figure (line 142).
- The authors have only shown the effects of MICTs on one AHR target gene (CYP1A1) (section Results 2.4). Did the authors investigate the effect of MICTs on other AHR target genes?
- What was the defined diet consumed by the subjects (line 220)? Was it supplemented with tryptophan? Is this defined diet a standard diet (line 289)? More clear details should be added.
- What do the authors mean by ‘After 24 h of stabilization, the cells…’ (line 385)? Did the authors starve the cells before the treatment with MICTs? If so, this information should be added (section Materials and Methods 4.4).
- For the cell viability assay, 10% DMSO is often used as a positive control for cell viability loss. Did the authors have used 10%DMSO or another experimental control?
- 'The cells were treated for 24 h with studied compounds in concentrations ranging from 1 nM to 200 μM (indole from 10 μM to 10 mM)’ (line 385-387). How did the authors define the studied compounds concentrations?
- Why two different luciferase assay systems were used? (line 400-401)
Minor issues:
- The image quality of Figure 1 is not good. The image resolution must be increased.
- The acronym ‘HTS’ should be added just after ‘High-throughput screening’ (line 146).
- The word ‘of’ should be removed in the sentence: ‘Heat-map of combination indexes (CI) values from high-throughput screening of the of MICT’ (line 184-185).

Author Response
RESPONSE TO THE REVIEWERS
REVIEWER #1
Aneta Vrzalova and coauthors investigated the effects of 11 tryptophan intestinal microbial metabolites (MICTs) on the aryl hydrocarbon receptor (AHR) activity. They analyzed the effect of individual, binary and mimic multicomponent mixtures of MICTs in order to get closer to the physiological human state. The authors identified synergistic as well as antagonistic effects of binary MICTs combinations on AHR activity which can be explained by a saturation of assay response. They also highlighted a determinative role of indole concentrations for gross AHR activity. Overall, the manuscript is clear, relevant for the field and presented in a well-structured manner. However, a series of issues need to be addressed:
Major issues:
Comment 1/1
It is not completely clear why the authors decided to analyze the effect of these 11 specific MICTs. I supposed that it is based on their recent study (Vyhlidalova et al., Trends Pharmacol Sci 2020) as outlined in the introduction section (line 79-89). The authors should clearly reformulate this part.
Response 1/1
The respective part of the manuscript was reformulated and the objectives of the study are now clearly defined in the revised manuscript.
Comment 1/2
Which cell line was used in Perdew’s lab study (line 85)? Is it a hepatic cell line as well? The authors should add this information as they set the incubation time regarding of this study.
Response 1/2
We thank the reviewer for this comment. The cell line used by Perdew´s Lab was hepatic reporter line Hepg2(40/6). This information was included in the revised manuscript.
Comment 1/3
The figure with the negative and positive controls for the luciferase assay should be added as a supplemental figure (line 142).
Response 1/3
The negative control is always per se “fold induction 1×”. The fold inductions of positive controls for each cell line and receptor type are now included in the revised manuscript, directly in Results, section 2.1.
Comment 1/4
The authors have only shown the effects of MICTs on one AHR target gene (CYP1A1) (section Results 2.4). Did the authors investigate the effect of MICTs on other AHR target genes?
Response 1/4
We have measured CYP1A1 mRNA levels as a proof of concept following the evaluation of binary mixtures in large-scale screening by reporter gene assay. The aim was to show that the results obtained by reporter gene assay are translated to and consistent with those obtained at mRNA level of the endogenous gene.
Comment 1/5
What was the defined diet consumed by the subjects (line 220)? Was it supplemented with tryptophan? Is this defined diet a standard diet (line 289)? More clear details should be added.
Response 1/5
(i) In a group “defined diet”, the stool samples were obtained from a controlled clinical nutrition diet study (Tindall et al 2019 J Am Heart Assoc 8:e011512), as described in Perdew´s paper (Dong et al 2020 Gut Microbes 12:e1788899). This information was included in the revised manuscript.
(ii) “standard diet” mentioned in line 289 refers to the “defined diet”. To avoid confusion, the word “standard” was replaced by “defined”, in the revised manuscript.
Comment 1/6
What do the authors mean by ‘After 24 h of stabilization, the cells…’ (line 385)? Did the authors starve the cells before the treatment with MICTs? If so, this information should be added (section Materials and Methods 4.4).
Response 1/6
The cells were let to stabilize for 24 h prior to the treatments. This time period allows the cells to properly attach to the culture surface and to refresh from stress caused by passaging and plating. To avoid confusion, we removed the word “stabilize” from the revised manuscript.
Comment 1/7
For the cell viability assay, 10% DMSO is often used as a positive control for cell viability loss. Did the authors have used 10%DMSO or another experimental control?
Response 1/7
We used 0.1% DMSO as negative control (vehicle), and Triton X-100 as a positive control. We included this information in the revised manuscript.
Comment 1/8
'The cells were treated for 24 h with studied compounds in concentrations ranging from 1 nM to 200 μM (indole from 10 μM to 10 mM)’ (line 385-387). How did the authors define the studied compounds concentrations?
Response 1/8
The top concentration of tested compounds was adjusted based on their solubility. This information was included in the revised manuscript.
Comment 1/9
Why two different luciferase assay systems were used? (line 400-401)
Response 1/9
We used six different stable reporter cell lines, which expressed different types of luciferase. The construction and validation of all cell lines were described and cited in the paper. The more recently developed cell lines PAZ-PPARg and IZ-VDRE contain an advanced NanoLuc® luciferase assay system, which is more sensitive than firefly luciferase assay system, used in other cells lines. The type of luciferase is irrelevant to the assay performance in relation to the current study.
Comment 1/10
Minor issues:
- The image quality of Figure 1 is not good. The image resolution must be increased.
- The acronym ‘HTS’ should be added just after ‘High-throughput screening’ (line 146).
- The word ‘of’ should be removed in the sentence: ‘Heat-map of combination indexes (CI) values from high-throughput screening of the of MICT’ (line 184-185).
Response 1/10
Corrections were done according to the reviewer’s comments.
REVIEWER #2
The authors of the manuscript entitled “Mixture effects of tryptophan intestinal microbial metabolites 2 on aryl hydrocarbon receptor activity” have described a Aryl hydrocarbon receptor responsiveness did not correlate with the type of diet or health status, and these were indole concentrations in the mixtures that were determinative for gross Aryl hydrocarbon receptor activity.
Strengths of the study:
- This manuscript has been written in well-structured that is collected through the previously published paper on a relevant topic.
- The reference list is updated, but there is a need to add some more references.
Comments:
- In the introduction part, is well structured but should be defined your study elaborately.
- should be focused on the English language with minor grammatical errors.
To conclude, this paper is a nice compact description of the study of the effects of tryptophan intestinal microbial metabolites on aryl hydrocarbon receptor activity.
This paper is accepted after minor improvements in introductions and grammatical errors.
Comment 2/1
In the introduction part, is well structured but should be defined your study elaborately.
Response 2/1
The respective part of the manuscript was reformulated and the objectives of the study are now clearly defined in the revised manuscript. See also Response 1/1.
Comment 2/2
Should be focused on the English language with minor grammatical errors.
Response 2/2
The English language was checked by the native speaker.
Reviewer 2 Report
The authors of the manuscript entitled “Mixture effects of tryptophan intestinal microbial metabolites 2 on aryl hydrocarbon receptor activity” have described a Aryl hydrocarbon receptor responsiveness did not correlate with the type of diet or health status, and these were indole concentrations in the mixtures that were determinative for gross Aryl hydrocarbon receptor activity.
Strengths of the study:
- This manuscript has been written in well-structured that is collected through the previously published paper on a relevant topic.
- The reference list is updated, but there is a need to add some more references.
Comments:
- In the introduction part, is well structured but should be defined your study elaborately.
- should be focused on the English language with minor grammatical errors.
To conclude, this paper is a nice compact description of the study of the effects of tryptophan intestinal microbial metabolites on aryl hydrocarbon receptor activity.
This paper is accepted after minor improvements in introductions and grammatical errors.
Author Response

(The authors gave the same response as above.)
